# The Usefulness of the C_2_HEST Score in Predicting the Clinical Outcomes of COVID-19 in COPD and Non-COPD Cohorts

**DOI:** 10.3390/microorganisms12061238

**Published:** 2024-06-20

**Authors:** Jakub Gawryś, Adrian Doroszko, Olgierd Dróżdż, Małgorzata Trocha, Damian Gajecki, Karolina Gawryś, Ewa Szahidewicz-Krupska, Maciej Rabczyński, Krzysztof Kujawa, Piotr Rola, Agata Stanek, Janusz Sokołowski, Marcin Madziarski, Ewa Anita Jankowska, Agnieszka Bronowicka-Szydełko, Dorota Bednarska-Chabowska, Edwin Kuźnik, Katarzyna Madziarska

**Affiliations:** 1Clinical Department of Internal and Occupational Diseases, Faculty of Medicine, Hypertension and Clinical Oncology, Wroclaw Medical University, Borowska Str. 213, 50-556 Wroclaw, Poland; jakub.gawrys@umw.edu.pl (J.G.); karolina.gawrys@umw.edu.pl (K.G.); ekrupska@gmail.com (E.S.-K.); 2Clinical Department of Cardiology, 4th Military Hospital, Faculty of Medicine, Wroclaw University of Science and Technology, Weigla 5 Str., 50-981 Wroclaw, Poland; adrian.doroszko@pwr.edu.pl (A.D.); damian.gajecki@pwr.edu.pl (D.G.); 3Clinical Department of Diabetology and Internal Diseases, Faculty of Medicine, Wroclaw Medical University, Borowska Str. 213, 50-556 Wroclaw, Poland; malgorzata.trocha@umw.edu.pl (M.T.); maciej.rabczynski@umw.edu.pl (M.R.); dorota.bednarska-chabowska@umw.edu.pl (D.B.-C.); edwin.kuznik@usk.wroc.pl (E.K.); katarzyna.madziarska@umw.edu.pl (K.M.); 4Statistical Analysis Centre, Wroclaw Medical University, K. Marcinkowski Str. 2-6, 50-368 Wroclaw, Poland; krzysztof.kujawa@umw.edu.pl; 5Department of Cardiology, Provincial Specialized Hospital, Iwaszkiewicz Str. 5, 59-220 Legnica, Poland; piotr.rola@gmail.com; 6Department and Clinic of Internal Medicine, Angiology and Physical Medicine, Faculty of Medical Sciences in Zabrze, Medical University of Silesia, Batory Str. 15, 41-902 Bytom, Poland; astanek@tlen.pl; 7Department of Emergency Medicine, Faculty of Medicine, Wroclaw Medical University, Borowska Str. 213, 50-556 Wroclaw, Poland; janusz.sokolowski@umw.edu.pl; 8Clinical Department of Rheumatology and Internal Medicine, Faculty of Medicine, Wroclaw Medical University, Borowska Street 213, 50-556 Wroclaw, Poland; marcin.madziarski@usk.wroc.pl; 9Institute of Heart Diseases, Faculty of Medicine, Wroclaw Medical University, Borowska Street 213, 50-556 Wroclaw, Poland; ewa.jankowska@umw.edu.pl; 10Department of Biochemistry and Immunochemistry, Faculty of Medicine, Wroclaw Medical University, Chałubińskiego St.10, 50-368 Wrocław, Poland; agnieszka.bronowicka-szydelko@umw.edu.pl

**Keywords:** COVID-19, SARS-CoV-2, COPD, prognosis, C_2_HEST score

## Abstract

Patients with chronic obstructive pulmonary disease (COPD) infected with SARS-CoV-2 indicate a higher risk of severe COVID-19 course, which is defined as the need for hospitalization in the intensive care unit, mechanical ventilation, or death. However, simple tools to stratify the risk in patients with COPD suffering from COVID-19 are lacking. The current study aimed to evaluate the predictive value of the C_2_HEST score in patients with COPD. A retrospective analysis of medical records from 2184 patients hospitalized with COVID-19 at the University Hospital in Wroclaw from February 2020 to June 2021, which was previously used in earlier studies, assessed outcomes such as mortality during hospitalization, all-cause mortality at 3 and 6 months, non-fatal discharge, as well as adverse clinical incidents. This re-analysis specifically examines the outcomes using a COPD split. In the COPD group, 42 deaths were recorded, including 18 in-hospital deaths. In-hospital mortality rates at 3 and 6 months did not significantly differ among C_2_HEST strata, nor did their impact on subsequent treatment. However, a notable association between the C_2_HEST score and prognosis was observed in the non-COPD cohort comprising 2109 patients. The C_2_HEST score’s predictive ability is notably lower in COPD patients compared to non-COPD subjects, with COPD itself indicating a high mortality risk. However, C_2_HEST effectively identifies patients at high risk of cardiac complications during COVID-19, especially in non-COPD cases.

## 1. Introduction

Despite restrictions, vaccination, and increasing knowledge regarding SARS-CoV-2 infection, COVID-19 remains one of the leading causes of hospitalization and patient deaths worldwide within the last three years. According to the data from Johns Hopkins University of Medicine on 3 October 2023, there have been more than 676 million infections since the beginning of the pandemic, of which almost 6.9 million have ended in death [1]. As the course of the disease differs significantly depending on general health, age, and comorbidities, it seems necessary to create or adapt already existing validated scales to implement them to precisely identify individuals at risk of developing severe SARS-CoV-2 infection. Besides the respiratory failure observed during the initial phase of the pandemic, we now know that SARS-CoV-2 infection increases the risk of developing cardiovascular, gastrointestinal, neurological, and rheumatic diseases, which are collectively referred to as “long COVID” [2].

Patients with chronic respiratory diseases, in particular chronic obstructive pulmonary disease (COPD), are at higher risk of developing a severe course of COVID-19. This higher risk may be related to decreased respiratory reserve, increased expression of angiotensin-converting enzyme-2 (ACE-2) receptor in the lower respiratory tract, which facilitates viral endocytosis, or impaired innate immunity mechanisms associated with prolonged exposure to the pathogen [3,4]. Meta-analyses indicate a higher risk of severe disease, which is defined as the need for hospitalization in the intensive care unit, mechanical ventilation, or death in COPD patients infected with SARS-CoV-2 [5,6]. Despite that, the prevalence of patients with COPD among hospitalizations due to SARS-CoV-2 infection ranges, according to available data, from 0.8% to 14.4%; however, among people hospitalized in the ICU, the rate is only 4% with approximately 12% prevalence in the general population [4,7].

The C_2_HEST score was initially designed to predict the risk of atrial fibrillation (AF); due to its simplicity, it offers an interesting value as a helpful tool for identifying the risk of the severe clinical course of COVID-19. Its components are based on the comorbidities that have been previously shown to have a major impact on the severity of the disease and prediction of critical illness [8]. Hence, the C_2_HEST score might be a useful tool in predicting the prognosis of patients with concomitant COPD suffering from SARS-CoV-2 infection. 

As the usefulness of this score has been previously shown to predict COVID-19 outcomes (death, non-fatal clinical events, ICU hospitalizations) in the diabetic population and people with heart failure in our previous studies, this study verifies its prognostic efficacy in the COPD and non-COPD cohorts [9,10].

## 2. Materials and Methods

### 2.1. Study Design and Participants

A retrospective analysis of the hospitalization records of COVID-19 patients at the Wroclaw Medical University Center from February 2020 to June 2021 was performed. The study design and overall study group were published by Rola et al. (2022); the work presented here is a re-analysis of their published data [9]. For the purposes of this research, a subpopulation of individuals with COPD was selected from the given cohort. The COLOS retrospective study protocol received approval from the Institutional Review Board and Ethics Committee of Wroclaw Medical University, Wroclaw, Poland (No: KB-444/2021). The records used for the analyses were taken from the routine retrospective results; therefore, informed consent was not necessary to be obtained from the participants. All study participants had a confirmed SARS-CoV-2 infection determined by nasopharyngeal swab reverse transcription polymerase chain reaction test (RT-PCR). The diagnosis of COPD was confirmed based on their medical history according to the Global Initiative for Chronic Obstructive Lung Disease (GOLD) 2023 criteria [11]. The database used for analyses included demographic information; the need for oxygen support therapy and its type; tobacco smoking history; concomitant diseases; medical pharmacological treatment before admission; laboratory findings; and the adverse events occurring during hospitalization including septic shock, venous thromboembolism (deep vein thrombosis or pulmonary embolism), acute heart failure, acute kidney injury, acute liver dysfunction, ARDS, and bleedings.

### 2.2. Follow-Up and Outcomes

The follow-up period started on the day of in-hospital admission and ended on the day of the patient’s discharge or death. Further information regarding the patients’ deaths was collected 3 and 6 months after admission. The individual clinical records were used to obtain patients’ characteristics. The collected details included the in-hospital death rate; mortality at 3 and 6 months regardless of cause; and the nature of hospital discharge not involving death (home discharge, relocation to a different hospital, or need for stationary rehabilitation).

### 2.3. C_2_HEST Score Stratification

In this study, a total of 75 consecutive patients with COPD and 2109 patients without a diagnosis of COPD assigned to the control group were included. The baseline characteristics of participating subjects were taken from the database and used to calculate the C_2_HEST score with a total of 6 individual components Patients could receive 1 point each for the presence of coronary artery disease, COPD in the medical history, arterial hypertension, and prior thyroid diseases. Characteristics such as age over 75 years and systolic cardiac failure earned the participant 2 points on the scale. Significantly, the requirement for coronary artery disease was fulfilled by a past occurrence of either myocardial infarction or coronary revascularization (MI, counted as 1 point). Furthermore, in subsequent sensitivity assessments, the term “thyroid disease” was substituted with greater specificity, delineating between “hyperthyroidism” and “hypothyroidism”. These risk elements were established through a blend of examining medical records and conducting interviews during initial visits. Afterward, according to the scale creators, the patients were divided into low (0–1 points), medium (2–3 points), and high (4 points and more) primary risk categories.

### 2.4. Statistical Analysis

The presentation of descriptive data includes categorical values expressed in percentages, while numerical variables are represented by the mean, standard deviation, and distribution of results (minimum to maximum) along with the count of available data points. For comprehensive testing, a chi-square test was employed for categorical variables having over 5 anticipated instances in every group, whereas Fisher’s exact test was utilized for fewer instances. Welch’s ANOVA was conducted for continuous variables due to unequal variances between the risk strata and a sufficiently large sample to obtain adequate asymptotic results. The Games–Howell test with Tukey correction was used for post hoc analysis for continuous variables. For categorical variables, a follow-up test mirrored the omnibus test, albeit conducted within subgroups and adjusted with Bonferroni correction. Hospital death and mortality from all causes were presented as data that were censored from the right, prompting the utilization of time-dependent ROC analysis coupled with an inverse probability of censoring weighting (IPCW) estimation for these variables. The C_2_HEST score was acquired through the utilization of the time-varying area under the curve (AUC). A log-rank test was used to verify the differences in survival curves among the various risk groups. The Grambsch–Therneau test was employed to confirm the assumption of proportional hazards. A Cox proportional hazards model was applied to examine the hazard ratio (HR) for the overall C_2_HEST score, its individual components, and the various risk categories. For the secondary outcomes, given their binary nature, a logistic regression model was applied. For assessing the predictive capabilities, the classical receiver operating characteristic (ROC) analysis and the AUC measure were used. An odds ratio (OR) was reported as the measure of effect for the influence of the C_2_HEST score, its individual components, and the different risk categories. All statistical analyses were conducted using R version 4.0.4, with the packages timeROC, pROC, survival, coin, and odds ratio [12,13,14,15]. A threshold of 0.05 was chosen for determining statistical significance in all analyses.

## 3. Results

### 3.1. Initial Characteristics and Comorbidities of the Study Population

The study and control group baseline characteristics are presented in Table 1. In the control group, a higher C_2_HEST score was associated with more advanced age, male sex, cigarette smoking, and the number of comorbidities. Moreover, the prevalence of almost all considered comorbidities (except for asthma) and tobacco smoking was significantly elevated in the group with the higher risk category according to the C_2_HEST scale. In the study group, the C_2_HEST score was related to higher average age, incidence of hypertension, atrial arrhythmia, heart failure, and history of myocardial ischemia.

From the patient-reported symptoms and vital signs of the non-COPD group, a higher C_2_HEST score was associated with a greater incidence of cough, dyspnea, and smell dysfunction, and differences in body temperature, heart rate, systolic blood pressure, pulse pressure, and lower baseline blood saturation measured by pulsoxymetry. In the physical examination, the prevalence of crackles, wheezing, pulmonary congestion, and peripheral edema was also higher in this cohort. In the COPD group, there were no variations in the vital signs, symptoms, and physical examination findings upon hospital admission among the strata based on the C_2_HEST score. Interestingly, except for dyspnea and blood saturation, significant disparities in terms of cough, chest pain, taste impairment, and other assessed vital signs were not detected between the COPD and non-COPD cohorts (Table 2).

### 3.2. Characteristics of the In-Hospital Laboratory Tests and Treatment Applied 

#### 3.2.1. Laboratory Assays

The full available characteristics of the laboratory results from the period of hospitalization for the COPD and non-COPD groups are summarized in the Appendix A. The measured parameters include routine blood count, blood gas analysis, selected biochemical parameters (including IL-6, vitamin B12, and iron metabolism), blood coagulation, and concentration of selected hormones (TSH, cortisol, and parathormone). Both groups are characterized by lower hemoglobin levels in the high-risk C_2_HEST strata. In the control group, there was a noticeable difference in the distribution of leukocytes (lymphopenia and neutropenia) and the prevalence of thrombocytopenia at the end of hospitalization among the C_2_HEST strata. During the whole hospitalization period, the high-risk category was characterized by higher lactate, ALAT, and TSH levels on admission and lower CRP levels upon discharge. The D-dimers, INR, APTT, glucose, urea, creatinine, albumin/protein, bilirubin, troponin, LDL, TG, and BNP values remained elevated throughout the whole hospitalization period. Among patients diagnosed with COPD, with the exception of a higher eosinophil count, lower procalcitonin concentration at discharge, and higher APTT, INR, TSH, glucose, urea, and creatinine levels on admission, there were no significant differences among the C_2_HEST strata.

#### 3.2.2. Treatment Applied during the Hospitalization Period 

There were no differences in the administration of COVID-19-specific treatment, including corticoids, remdesivir, tocilizumab, nor convalescent plasma among the C_2_HEST strata in both studied cohorts. Only the use of antimicrobial agents was higher in the high-risk stratum from the control group.

As for supportive treatment, in the control group, with increasing C_2_HEST score, there was an extended need for oxygen therapy; in addition, coronary artery revascularization and the use of catecholamines were more common among high-risk patients. Interestingly, in the COPD cohort, no differences regarding the above-mentioned parameters were found in the C_2_HEST strata (Table 3).

### 3.3. Associations of the C_2_HEST Score with Fatal Outcomes 

#### 3.3.1. C_2_HEST Score Results and Mortality

A total of 42 deaths (73.7%) from 75 patients were reported in the COPD cohort with 18 (24%) in-hospital events. The mortality rates during hospitalization at 3 and 6 months among the C_2_HEST strata as well as its influence on the type and destination of further treatment did not significantly differ. On the contrary, in the non-COPD cohort, a strong, significant association between the C_2_HEST score and the prognosis was observed (Table 4).

#### 3.3.2. Differentiating Ability of the C_2_HEST Score in Predicting Overall Mortality

The receiver operating characteristic (ROC) analysis revealed that C_2_HEST failed to predict the 1-, 3-, and 6-month mortality in the COPD cohort. On the other hand, it was revealed to be a fair indicator of death prevalence in the non-COPD group. The C_2_HEST predicting values assessed by the AUC analysis in the COPD vs. non-COPD cohorts are as follows: the 1-month AUC30 = 44.8% vs. 70.8%; 3-month AUC90 = 53.3% vs. 71.8%; and 6-month AUC180 = 56.1% vs. 70.0%. All data were computed considering all-cause mortality without competing risks (Figure 1).

In addition, a timeROC analysis was performed to assess the predictive ability of the C_2_HEST score for deaths after a specified time “t” from hospital admission. All causes of death were considered in the analysis. Figure 2 shows the predictive ability expressed as the area under the ROC curve as a function of time, together with the confidence intervals.

#### 3.3.3. Discriminatory Performance of the C_2_HEST Score on the In-Hospital All-Cause Mortality–Time–ROC

An analysis presented in Figure 3 shows the time-dependent AUC for the C_2_HEST score in predicting the in-hospital deaths in both cohorts. The predicting abilities, regardless of the time of hospitalization, were poor in the control group while it failed in the COPD cohort.

#### 3.3.4. The Probability of Survival in Hospitalized COVID-19 Patients

The Kaplan–Meier functions were used to estimate the survival curves for C_2_HEST groups according to the first categorization of low/medium/high for 0–1/2–3/≥4 points, respectively. The curves were compared using the Log-rank test—a *p*-value of 0.6604 indicates that the probability of survival in the risk strata for the COPD cohort is not statistically significant (Figure 4). On the contrary, in the non-COPD cohort, the *p*-value was <0.0001. 

#### 3.3.5. Risk Strata Matching for Analysis

To determine whether the original layout, which includes the low/medium/high-risk categories for 0–1/2–3/≥4 points, respectively, of the C_2_HEST scale is optimal for patients with COPD, an analysis was performed taking into consideration all the possible C_2_HEST intervals with the performance of the log-rank statistics test for each one (Appendix A). 

The highest value of the log-rank statistics for the COPD cohort corresponded with the C_2_HEST strata, which were estimated as follows:●0–4—low;●5–5—medium;●6–8—high.

This calculation results in better risk stratification than the generally accepted one; however, the original subdivision method is used in the following sections of this study. Similar statistical analysis regarding total and in-hospital mortality revealed that, for the non-COPD cohort, the main risk categories (0–1 = low; 2–3 = medium; and >4 = high) most accurately represent the mortality trends (Appendix A).

#### 3.3.6. Effect of the C_2_HEST Risk Stratification Result on COVID-19 Survival

A power analysis of the effect of scale results on survival was performed using the Cox model. It tested how increasing the scale score by 1 and how changing the risk group affects mortality. In the COPD and non-COPD groups, an increase in one point resulted in a death rate increase of 4.1% and 42.4%, respectively. Upgrading the risk category from low to medium changed the death likelihood by 0.84 and 3.41 as well as 1.15 and 5.11 between the low-risk and high-risk groups in the COPD and non-COPD cohorts (Table 5). Importantly, the results in the group of subjects with a diagnosis of COPD are not statistically significant (*p* > 0.05) in opposition to people without such a diagnosis. 

A similar analysis was performed for in-hospital deaths. For the COPD cohort, a one-point increase in C_2_HEST score was related to an 11.6% increase in in-hospital death. This risk does not significantly change between the medium- vs. low-risk and high- vs. low-risk strata. The same statistical model for the non-COPD cohort revealed a 28.3% increase in in-hospital mortality for each C_2_HEST point and a hazard ratio of 2.34 after the change from low- to medium-risk compartment, and 3.05 after the change from low- to high-risk, respectively (Table 6). 

The associations of individual C_2_HEST score components with mortality and other selected endpoints are included in the Appendix A. Cox model analysis showed the greatest effect on in-hospital mortality of age and diagnosis of heart failure with reduced ejection fraction for the group with COPD and age and diagnosis of coronary artery disease for the non-COPD cohort, respectively. 

#### 3.3.7. Associations of the C_2_HEST Score with Other, Non-Fatal Events 

The receiver operating characteristic (ROC) analysis performed on the COPD cohort revealed that the C_2_HEST predicts hypovolemic shock (AUC = 0.715), deep vein thrombosis (AUC = 0.959), acute heart failure (AUC = 0.847), multi-organ dysfunction syndrome (MODS) (AUC = 0.801), and all bleedings (AUC 0.701) including respiratory tract (AUC = 0.959), upper gastrointestinal tract (AUC = 0.808), and urinary tract (AUC = 0.797) bleeding. The Cox model showed that a 1-point change in this scale significantly predicts only the risk of acute heart failure (2.5-fold increase, 95% CI = 1.24–7.32, *p* = 0.0312). For a group without COPD, the ROC showed that C_2_HEST predicts cardiogenic shock (AUC = 0.773), acute heart failure (AUC = 0.869), and new cognitive signs and symptoms (AUC = 0.723). Each additional point in the C_2_HEST score increased the risk of cardiogenic shock by 72% with 4.6-fold (95% CI 1.65–13.80, *p* < 0.0001) and 12.8-fold (95% CI 4.96–13.80, *p* = 0.0039) increases for strata change from low- to medium-risk and low- to high-risk, respectively. For the acute heart failure, the risk increases by 112% with each C_2_HEST point (OR low vs. medium: 8.67, 95% CI = 3.99–20.90, *p* < 0.0001; OR low vs. high: 39.20, 95% CI = 19.12–91.26, *p* < 0.0001) and by 48% (OR low vs. medium: 4.56, 95% CI = 2.95–7.10, *p* < 0.0001; OR low vs. high: 5.36, 95% CI = 3.20–8.89, *p* < 0.0001) for the new cognitive signs and symptoms. The detailed results of the analyses can be found in Table 7.

For the non-COPD cohort, it is noteworthy that the C_2_HEST score can predict the occurrence of myocardial infarction and injury, stroke/TIA, acute kidney and liver dysfunction, and pneumonia. Surprisingly, there was no significant prognostic value for an occurrence of deep vein thrombosis, pulmonary embolism, and septic shock. With the exception of acute heart failure, this scale had no predictive value for adverse clinical outcomes in the COPD cohort.

#### 3.3.8. Sensitivity Analysis

It is noteworthy that when calculating the C_2_HEST score, taking into account “hypothyroidism” in the place of “thyroid disease” and altering the cut-off point for age from above 75 years to above 65 years led to a notable enhancement in the predictive efficacy of the tested scale in the control group, but it did not have an impact on its parameters in the COPD group regarding the primary study endpoints and most of the adverse clinical events, with the exception of worsening the predictive value for acute heart failure in study group. The findings from the sensitivity analysis are outlined in the Appendix A.

## 4. Discussion

To the best of our knowledge, this study is the initial examination of the prognostic utility of the C_2_HEST scale in predicting mortality risk in COPD patients with COVID-19. The usefulness of this scale in assessing the risk of death has been demonstrated in other cohorts selected from the COLOS study population, including subjects with type 2 diabetes or heart failure [9,10]. Reports from the beginning of the pandemic indicated the diagnosis of chronic respiratory diseases, in particular COPD, which together with active smoking significantly increased the risk of a severe course of SARS-CoV-2 infection, which is defined as the need for mechanical ventilation or death. According to a meta-analysis performed by Zhao et al., the risk of a severe course of COVID-19 among patients with COPD increased by 4.38 fold, with the prevalence of severe course of COVID-19 widely differing among the component studies (6.10–28%) [6]. In our study, which takes into account a longer period (17 months vs. 4 months), the mortality rate in the COPD cohort reached 24%, which is in the upper range of the initially analyzed studies [6]. Due to the substantial heterogeneity in the population diagnosed with COPD—from those requiring only low doses of inhalation drugs to those requiring continuous oxygen therapy at home—it appears necessary to find a tool to anticipate the patient’s outcome and to identify the group of patients requiring special attention.

The initial purpose of the C_2_HEST scale was to forecast the likelihood of atrial fibrillation (AF) occurrence [16]. Its predictive ability has been demonstrated in large population-based studies [17,18,19]. Some authors have proven its efficacy not only in AF prediction but also in anticipation of other clinical outcomes such as death or requirement for hospitalization among patients diagnosed with heart failure and preserved ejection fraction (HFpEF) [20]. Among the COLOS study population, this scale also showed a potent predictive value, not only for mortality and the risk for a severe course of COVID-19, but also for other endpoints such as the development of acute liver or renal failure, the incidence of shock, or the need for more aggressive oxygen therapy (unpublished data). Interestingly, when considering only the population with a diagnosis of COPD, statistically significant differences were only found regarding cardiac complications—the occurrence of a new myocardial infarction or increased risk of acute heart failure and cardiogenic shock for each additional point in the C_2_HEST score. The lack of association, e.g., with mortality, may be due to the fact that the diagnosis of COPD itself is a factor in the significantly increased risk of death in COVID-19, though, among other aspects, such as reduced respiratory reserve, bronchial hyperresponsiveness, and impaired non-specific immunity resulting in the development of severe pneumonia [5,21].

Additionally, the predictive value of this scale may be reduced by the fact that one of its components is a diagnosis of COPD. Therefore, the cut-off points would need to be modified as proposed in this study. However, such a change would affect the clarity and convenience of the scale, so statistical calculations were performed for its original version. Despite these limitations, a statistically significant association of the C_2_HEST scale with acute heart failure was observed. Both heart failure and COPD have common risk factors for their development [22]. Furthermore, in COPD patients, there is a 20–32% prevalence of HF, while in the HF population, the comorbidity with COPD reaches 10% [23]. These two conditions in part also share the pathogenesis including endothelial dysfunction, oxidative stress, and aforementioned chronic inflammation [24,25]. The C_2_HEST score may help to identify patients at higher risk of developing not only acute heart failure but also myocardial infarction during hospitalization due to COVID-19. The diagnosis of COPD itself is associated with a higher incidence of myocardial infarction compared to the general population [26]. In addition, this risk increases during further exacerbations [27,28]. As COVID-19 symptoms depend not only on the activity of the virus itself but also on an enhanced inflammatory response, SARS-CoV-2 pneumonia, itself an exacerbation of COPD, is a risk factor for myocardial infarction [29]. In the case of rapidly developing acute respiratory failure, especially during a pandemic with a significantly overloaded healthcare system, the traditional risk assessment is difficult to perform [30]. Our study showed that the C_2_HEST score, due to its simplicity, enables easy identification of patients with a particularly high risk for adverse outcomes.

## 5. Limitations

Our study has several limitations. Firstly, in the study cohort, the number of patients with COPD was relatively low, accounting for less than 5% of the entire study population. Moreover, our results are based on a retrospective analysis of cases of patients hospitalized in a single center, which could affect the validity of our conclusions. Furthermore, all patients included in this study were from Central and Eastern Europe, which may limit relating the obtained conclusions to the world population. Furthermore, the data of the presented study group were from the years 2020–2021, which is the early period of the pandemic when vaccines were either not yet available or accessible to only a small portion of the population. This means that the presented database does not include vaccination information, which could have impacted the results related to incidence, disease severity, and mortality due to COVID-19. Finally, this study pertains to the early years of the pandemic; new variants of the virus began to emerge in the subsequent years. Consequently, the results concerning the C_2_HEST scale from this study may not be replicable for new groups infected with different variants of the virus.

## 6. Conclusions

In conclusion, the C_2_HEST score enables the prediction of the risk of death and many other complications in the course of SARS-CoV-2 infection. Within the population of patients with COPD, the predictive ability of this scale is much lower; however, the C_2_HEST score easily identifies patients with a particularly high risk of cardiac complications in the course of COVID-19.

## Figures and Tables

**Figure 1 microorganisms-12-01238-f001:**
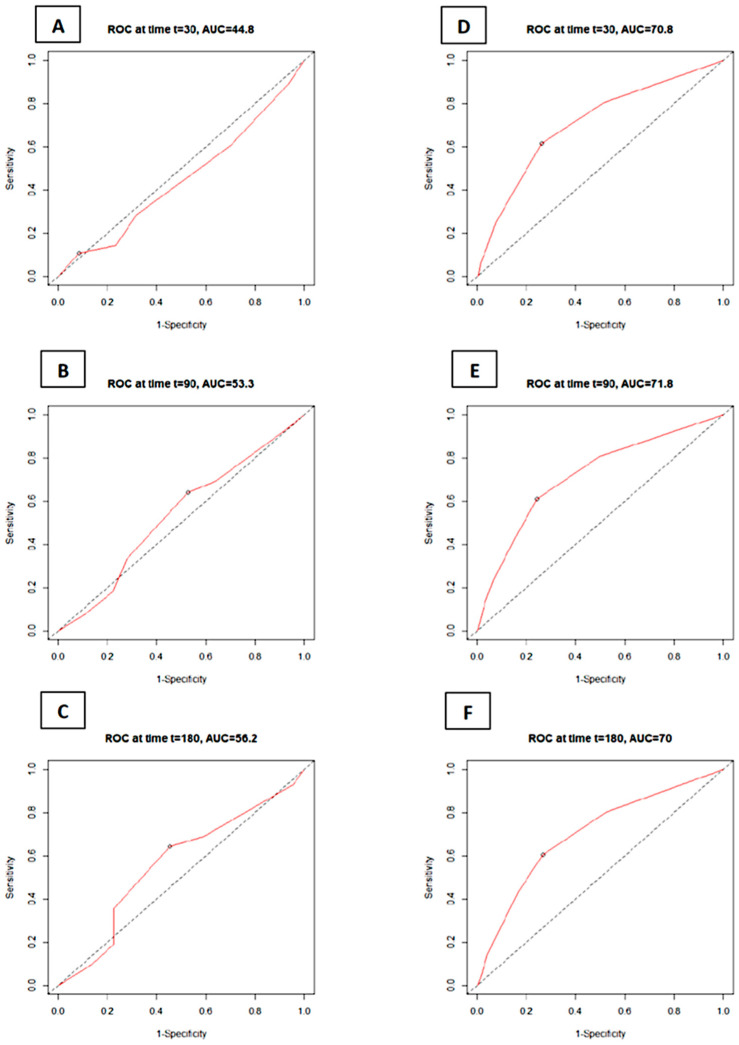
ROC curves for the C_2_HEST score in predicting total mortality in the study groups: COPD (**A**–**C**) and control non-COPD groups (**D**–**F**) at selected time points (30 days; 90 days; and 180 days) after the positive RT-PCR test. Abbreviations: area under the curve—AUC; receiver operating characteristic—ROC.

**Figure 2 microorganisms-12-01238-f002:**
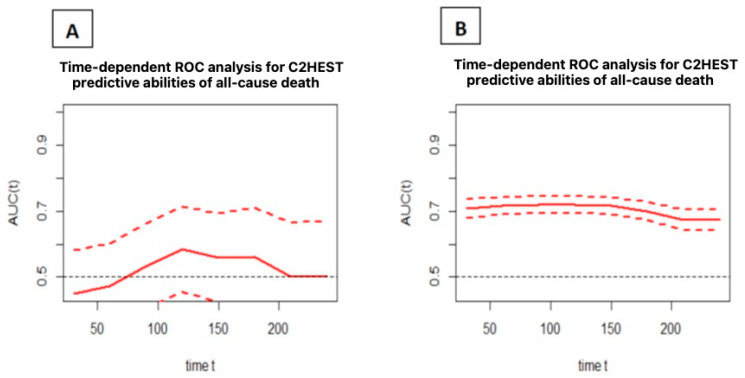
Time-dependent ROC analysis for the C_2_HEST predictive values of all-cause death in the study (**A**) and control (**B**) groups (mean with CI). Abbreviations: area under the curve—AUC; receiver operating characteristic—ROC.

**Figure 3 microorganisms-12-01238-f003:**
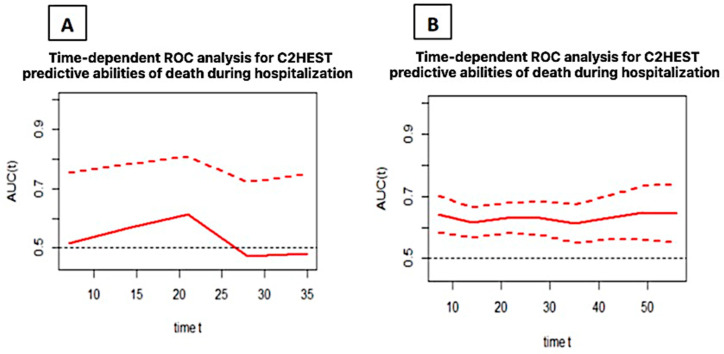
Time-dependent ROC analysis for the C_2_HEST predictive values of in-hospital all-cause death in the study (**A**) and control (**B**) groups (mean with CI). Abbreviations: area under the curve—AUC; receiver operating characteristic—ROC.

**Figure 4 microorganisms-12-01238-f004:**
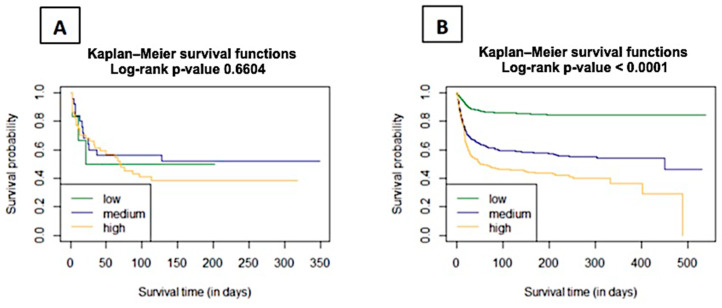
Analysis of 6-month survival across low, medium, and high C_2_HEST risk categories in the study (**A**) and control (**B**) groups (mean with CI). Abbreviation: area under the curve—AUC.

**Table 1 microorganisms-12-01238-t001:** The baseline demographic characteristics and concomitant diseases of the study and control groups.

Variables, Units	All Pts		Low Risk [0–1]		Medium Risk [2–3]		High Risk [≥4]		*p*-Value		Post Hoc Analysis for Significant *p*	
Demographics	No COPD n = 2109	COPD n = 75	No COPD n = 1412	COPD n = 6	No COPD n = 467	COPD n = 25	No COPD n = 230	COPD n = 44	No COPD	COPD	No COPD	COPD
Age, years mean ± SD n, (min.–max.)	59.6 ± 19.0 2109 (17–100)	72.0 ± 8.06 75 (54–94)	51 ± 15.9 1412 (17–74)	66.7 ± 5.6 6 (57–71)	76 ± 11.8 467 (29–100)	67.4 ± 5.7 25 (41–97)	79.2 ± 9.5 230 (38–100)	75.3 ± 7.9 44 (59–94)	<0.0001	<0.0001	<0.0001 ^a,b,c^	0.961 ^a^ 0.024 ^b^ <0.0001 ^c^
Age ≥ 65 years, n (%)	985 (46.7)	62 (82.7)	372 (73.7)	4 (66.7)	68 (14.6)	20 (80.0)	214 (93.0)	38 (86.4)	<0.0001	0.3111	<0.0001 ^a,b^ <0.0001 0.0004 ^c^	N/A
Male gender, n (%)	1033 (49.0)	49 (65.3)	730 (51.7)	5 (83.3)	192 (41.1)	16 (64.0)	111 (48.3)	28 (63.6)	0.0003	0.8078	0.0003 ^a^ 1 ^b^ 0.26 ^c^	
BMI, kg/m^2^ mean ± SD, (min.–max.),	28.4 ± 5.9 (15.4–49.4) 532	28.7 ± 4.5 22 (18.6–36.7)	28.3 ± 5.1 397 (15.4–49.4)	0	29.4 ± 5.7 81 (20.5–47.7)	28.4 ± 5.6 9 (18.6–36.7)	27.5 ± 6.3 54 (16.4–48.2)	28.8 ± 3.7 13 (22.9 –34.9)	0.07	0.8591		
Normal body weight (BMI = 18.5–24.9 kg/m^2^), n, n (%)	136 (25.6)	5 (22.7)	100 (25.1)	0	16 (19.6)	3 (33.3)	20 (37.0)	2 (15.4)				
Underweight (BMI < 18.5 kg/m^2^), n (%) n = 22	5 (1.0)	0 (0.0)	3 (0.76)	0 (0.0)	0 (0.0)	0 (0.0)	2 (3.7)	0 (0.0)				
Overweight (BMI = 25–29.9 kg/m^2^), n (%), n = 22	209 (39.3)	8 (36.4)	162 (40.8)	0	31 (38.3)	2 (22.2)	16 (29.6)	6 (46.2)				
Obesity (BMI ≥ 30 kg/m^2^), n (%), n = 22	182 (34.2)	9 (40.9)	132 (33.2)	0	34 (42.0)	4 (44.4)	16 (29.6)	5 (38.5)				
Tobacco smoking, never/previous/current, n (%), n = 75	1953/93/59 (92.8/4.4/2.8)	34/24/17 (45.3%/32%/22.7%)	1335/45/32 (94.5/3.2/2.3)	3/1/2 (50%/16.7%/33.3%)	421/26/17 (90.7/5.6/3.7)	10/9/6 (40%/36%/24%)	197/22/10 (86.0/9.6/4.4)	21/14/9 (47.7%/31.8%/20.5%)	<0.0001	0.8795	0.04 ^a^ <0.0001 0.38 ^c^	
Comorbidities												
Hypertension, n (%) n = 75	962 (45.6)	60 (80.0)	416 (29.5)	0	335 (71.7)	22 (88.0)	211 (91.7)	38 (86.4)	<0.0001	<0.0001	<0.0001 ^a,b,c^	<0.0001 ^a,b^ 1.0 ^c^
Diabetes mellitus, n (%), n = 473	486 (34.1)	30 (40.0)	234 (16.6)	1 (16.7)	144 (30.9)	10 (40.0)	108 (47.2)	19 (43.2)	<0.0001	0.9498	<0.0001 ^a,b^ 0.0002 ^c^	
Dyslipidemia, n (%) n = 31	574 (73.8)	15 (48.4) n = 31	288 (69.4)	1 (33.3) n = 3	161 (74.2)	7 (63.6) n = 11	125 (85.6)	7 (41.2) n = 17	0.0006	0.4939	0.001 ^a^ 0.72 ^b^ 0.04 ^c^	
Atrial fibrillation/flutter, n (%), n = 75	260 (12.3)	30 (40.0)	49 (3.4)	0	100 (21.4)	6 (24)	111 (48.2)	24 (54.5)	<0.0001	<0.0001	<0.0001 ^a,b,c^	0.928 ^a^ 0.0689 ^b^ 0.06698 ^c^
Previous coronary revascularization, n (%), n = 75	136 (6.4)	18 (24)	6 (0.4)	0	36 (7.7)	1 (4.0)	94 (40.9)	17 (38.6)	<0.0001	<0.0001	<0.0001 ^a,b,c^	1 0.295 ^b^ <0.0001 ^c^
Previous myocardial infarction, n (%), n = 75	170 (8.0)	21 (28.0)	11 (0.8)	0	60 (12.8)	3 (12.0)	99 (43.0)	18 (40.9)	<0.0001	<0.0001	<0.0001 ^a. b. c^	1 ^a^ 0.2263 ^b^ 0.044
Heart failure, n (%) n = 75	226 (10.7)	29 (38.7)	0 (0.0)	0	53 (11.34)	0	173 (75.2)	29 (65.9)	<0.0001	<0.0001	<0.0001 ^a,b,c^	1 ^a^ 0.0102 ^b^ <0.0001 ^c^
Moderate/severe valvular heart disease or previous valve heart surgery, n (%) n = 75	86 (4.1)	10 (13.3)	13 (0.9)	0	30 (6.4)	2 (8.0)	43 (18.7)	8 (18.2)	<0.0001	0.4162	<0.0001 ^a,b,c^	
Peripheral artery disease, n (%) n = 75	94 (4.5)	6 (8.0)	26 (1.8)	0	30 (6.4)	1 (4.0)	38 (16.5)	5 (11.4)	<0.0001	0.6465	<0.0001 ^a,b,c^	
Previous stroke/TIA, n (%), n = 75	151 (7.15)	13 (17.3)	47 (3.3)	0	57 (12.2)	2 (8.0)	47 (20.4)	11 (25)	0.0012	0.1318	<0.0001 ^a,b,c^	
Chronic kidney disease n = 75	212 (10.1)	19 (25.3)	69 (4.9)	1 (16.6)	66 (14.1)	4 (16.0)	77 (33.5)	14 (31.8)	<0.0001	0.3665	<0.0001 ^a,b,c^	
Hemodialysis, n (%) n = 75	53 (2.5)	5 (6.7)	19 (1.3	0	18 (3.9)	2 (8.0)	16 (7.0)	3 (6.8)	<0.0001	1	0.004 ^a^ <0.0001 ^b^ 0.033 ^c^	
Asthma, n (%) n = 75	77 (3.7)	8 (10.7)	54 (3.8)	0	17 (3.6)	3 (12.0)	6 (2.6)	5 (11.4)	0.66	1		
Thyroid disease, none/hypothyroidism/hyperthyroidism, n (%) n = 75	1890/199/20 (89.6/9.4/1.0)	65/9/1 (86.7%/12.0%/1.3%)	1332/76/4 (94.3/5.4/0.3)	6/0/0 (100.0%/0.0%/0.0%)	391–66–10 (83.7/14.1/2.2)	23/2/0 (92.0%/8.0%/0.0%)	167/57/6 (72.6/24.8/2.6)	36/7/1 (81.8%/15.9%/2.3%)	<0.0001	0.7322	<0.0001 ^a,b^ 0.0059 ^c^	

Presentation of variables: continuous mean ± standard deviation, results distribution (min.–max.), and the number of present values; the values are presented as numbers with a percentage. Information regarding the numbers showing the correct values can be found in the left column. List of abbreviations used: valid measurements—N; the count of patients exceeding the cut-off point—n; standard deviation—SD; body mass index—BMI; transient ischemic attack—TIA; chronic obstructive pulmonary disease—COPD; not applicable—N/A; low- vs. medium-risk groups—^a^; low- vs. high-risk groups—^b^; and medium- vs. high-risk groups—^c^.

**Table 2 microorganisms-12-01238-t002:** The symptoms reported by patients, measured vital signs, and baseline physical examination findings in the study and control groups.

Variables, Units	All Pts		Low Risk [0–1]		Medium [2–3]		High Risk [≥4]		*p*-Value		Post Hoc Analysis for Significant *p*	
Patient-Reported Symptoms	No COPD n = 2109	COPD n = 75	No COPD n = 1412	COPD n = 6	No COPD n = 467	COPD n = 25	No COPD n = 230	COPD n = 44	No COPD	COPD	No COPD	COPD
Cough, n (%) n = 75	628 (29.8)	20 (26.7)	455 (32.2)	0	116 (24.8)	8 (32.0)	57 (24.8)	12 (27.3)	0.0022	0.3753	0.01 ^a^ 0.09 ^b^ 1 ^c^	
Dyspnea, n (%) n = 75	869 (41.2)	52 (69.3)	567 (40.2)	2 (33.3)	190 (40.7)	16 (64.0)	112 (48.7)	34 (77.3)	0.0492	0.0606	1 ^a^ 0.053 ^b^ 0.16 ^c^	
Chest pain, n (%) n = 75	153 (7.3)	10 (13.3)	101 (7.2)	1 (16.7)	30 (6.4)	4 (16.0)	22 (9.6)	5 (11.4)	0.31	0.6684		
Hemoptysis, n (%) n = 75	15 (0.7)	0	9 (0.6)	0	2 (0.4)	0	4 (1.7)	0	0.13	<0.0001		0.002 ^a^ <0.0001 ^b^ 0.07 ^c^
Smell dysfunction, n (%), n = 75	75 (3.6)	1 (1.3)	61 (4.3)	0	10 (2.1)	0	4 (1.7)	1 (1.3)	0.025	1	0.13 ^a^ 0.18 ^b^ 1 ^c^	
Taste dysfunction, n (%), n = 75	64 (3.0)	2 (2.7)	49 (3.5)	0	10 (2.1)	0	5 (2.2)	2 4.5	0.25	0.6036		
Abdominal pain, n (%), n = 75	142 (6.7)	5 (6.7)	103 (7.3)	1 (16.7)	26 (5.6)	0	13 (5.7)	4 (9.1)	0.34	0.1704		
Diarrhea, n (%) n = 75	120 (5.7)	7 (9.3)	74 (5.2)	1 (16.7)	32 (6.9)	1 (4.0)	14 (6.1)	5 (11.4)	0.41	0.2900		
Nausea/Vomiting, n (%), n = 75	97 (4.6)	1 (1.3)	57 (4.0)	0	27 (5.8)	0	13 (5.7)	1 (2.3)	0.21	1		
**Measured vital signs**												
Body temperature, °C, mean ± SD (min.–max.), n = 40	37.0 ± 0.88 (34.3–40.5)	37.1 ± 0.97 (35.2–39.0) n = 40	37.1 ± 0.9 (34.4–40.5)	37.3 ± 1.58 (35.9–39.0) n = 3	36.9 ± 0.9 (35–40)	37.0 ± 0.85 (36.0–38.5) n = 13	36.9 ± 0.8 (35.5–40)	37.1 ± 0.99 (35.2–39.0) n = 24	0.032	0.9644	0.1 ^a^ 0.13 ^b^ 0.98 ^c^	
Heart rate, beats/minute mean ± SD (min.–max.), n = 66	85.6 ± 16.2 (36–160)	86.1 ± 19.69 (60–170) n = 66	86.4 ± 15.6 (48–160)	87.5 ± 23.63 (70–120) n = 4	83.9 ± 16.5 (50–160)	88.5 ± 16.02 (69–121) n = 20	84.7 ± 18.3 (36–150)	84.9 ± 21.22 (60–170) n = 42	0.03	0.7849	0.03 ^a^ 0.45 ^b^ 0.85 ^c^	
Respiratory rate, breaths/minute mean ± SD (min.–max.), n = 15	18.4 ± 5.6 (12–50)	21.4 ± 8.26 (16–50) n = 15	18.4 ± 5.8 (12–50)		18.5 ± 5.5 (12–45)	21.8 ± 2.36 (20–25) n = 4	18.7 ± 4.5 (12–30)	21.3 ± 9.69 (16–50) n = 11	0.92	0.882		
Systolic blood pressure, mmHg mean ± SD (min.–max.), n = 65	132.0 ± 22.9 (50–270)	134.2 ± 22.3 (85–184) n = 65	130.7 ± 21.3 (60–240)	131.0 ± 25.1 (105–155) n = 3	134 ± 25.6 (50–270)	138.6 ± 19.73 (90–167) n = 20	134.9 ± 25.0 (70–210)	132.4 ± 23.48 (85–184) n = 42	0.014	0.5991	0.07 ^a,b^ 0.9 ^c^	
Systolic blood pressure <100 mmHg, n (%), n = 65	73 (4.6)	5 (7.69) n = 65	45 (4.3)	0 (0.0) n = 3	17 (4.7)	1 (5.0) n = 20	11 (5.4)	4 (9.5) n = 42	0.78	0.9999		
Diastolic blood pressure, mmHg mean ± SD (min.–max.), n = 65	78.1 ± 13.4 (40–157)	76.9 ± 13.1 (45–110) n = 65	78.5 ± 12.7 (40–150)	80.0 ± 17.32 (70–100) n = 3	78.0 ± 13.8 (40–157)	79.0 ± 11.0 (50–95) n = 20	75.8 ± 15.6 (40–143)	75.7 ± 13.9 (45–110) n = 42	0.06	0.6501		
Mean blood pressure, mmHg MAP mean ± SD (min.–max.), n = 65	96.2 ± 14.9 (46.7–190)	96.0 ± 14.6 (58.3–125) n = 65	96.0 ± 14.2 (46.7–179)	97.0 ± 19.06 (81.7–118.3) n = 3	97.1 ± 15.6 (59.7–190)	98.8 ± 13.2 (63.3–115.7) n = 20	95.5 ± 17.1 (50–165.3)	94.6 ± 15.11 (58.3–125) n = 42	0.43	0.6001		
Pulse pressure mean ± SD (min.–max.), n = 65	54.3 ± 16.8 (11–136)	57.3 ± 17.3 (24–99) n = 65	52.3 ± 15.3 (11–136)	51.0 ± 14.4 (35–63) n = 3	57.1 ± 18.7 (50–100)	59.7 ± 12.8 (40–80) n = 20	59.2 ± 8.5 (20–130)	56.6 ± 19.4 (24–99) n = 42	<0.0001	0.6016	<0.0001 ^a,b^ 0.412 ^c^	
SpO_2_ on room air, % (FiO_2_ = 21%) mean ± SD (min.–max.), n = 52	91.2 ± 8.0 (48–100)	87.8 ± 9.18 (56–99) n = 52	92.9 ± 7.1 (48–100)	91.5 ± 5.69 (85–98) n = 4	89.9 ± 9.5 (50–100)	85.0 ± 11.29 (56–99) n = 14	90.6 ± 8.5 (50–99)	88.6 ± 8.45 (65–99) n = 34	<0.0001	0.3581	<0.0001 ^a^ 0.012 ^b^ 0.772 ^c^	
Sp O_2_ < 90%, n (%) n = 52	316 (15.0)	26 (50.0) n = 52	181 (22.3)	2 (50.0) n = 4	92 (34.5)	10 (71.4) n = 14	43 (32.3)	14 (41.2) n = 34	0.0001	0.2119	0.0003 ^a^ 0.048 ^b^ 1 ^c^	
GCS, points, n = 18	14.5 ± 1.9 (1–15)	14.8 ± 0.51 (13–15) n = 18	14.6 ± 1.8 (1–15)	0	14.5 ± 1.7 (3–15)	15.0 ± 0.0 (15–15) n = 6	14.1 ± 2.5 (3–15)	14.75 ± 0.62 (13–15) n = 12	0.049	NaN	0.38 ^a^ 0.07 ^b^ 0.34 ^c^	
**Abnormalities detected during physical examination**												
Crackles, n (%) n = 75 validated	304 (14.4)	15 (20.0)	153 (10.8)	1 (16.7)	96 (20.6)	3 (12)	55 (23.9)	11 (25.0)	<0.0001	0.4534	<0.0001 ^a,b^ 1 ^c^	
Wheezing, n (%) n = 75	187 (8.9)	32 (42.7)	92 (6.5)	2 (33.3)	49 (10.5)	7 (28.0)	46 (20.0)	23 (52.3)	<0.0001	0.1553	0.02 ^a^ <0.0001 ^b^ 0.003	
Pulmonary congestion, n (%) n = 75	343 (16.3)	24 (32)	183 (13.0)	1 (16.7)	98 (21.0)	7 (28.0)	62 (27.9)	16 (36.4)	<0.0001	0.6894	0.0001 ^a,b^ 0.29 ^c^	
Peripheral edema, n (%) n = 75	177 (8.4)	12 (16.0)	75 (5.3)	1 (16.7)	56 (12.0)	4 (16.0)	46 (20.0)	7 (15.9)	<0.0001	1	<0.0001 ^a,b^ 0.02 ^c^	
Hemiplegia/ hemiparesis, n (%) n = 75	70 (3.3)	3 (4.0)	31 (2.2)	0	24 (5.1)	0	15 (6.5)	3 (6.8)	0.0001	0.6498	0.006 ^a^ 0.002 ^b^ 1 ^c^	
VES–13, points n = 9	5.4 ± 3.2 (1–13)	5.1 ± 3.1 (1–10) n = 9	4.1 ± 2.9 (1–9)	0	5.8 ± 3.3 (1–12_	4 ± 4.36 (1–9) n = 3	6.7 ± 3.0 (3–13)	5.7 ± 2.58 (3–10) n = 6	0.13	0.5884	0.094 ^a^ 0.014 ^b^ 0.542 ^c^	

Presentation of variables: continuous mean ± standard deviation, results distribution (min.–max.), and the number of present values; the values are presented as numbers with a percentage. Information regarding the numbers showing the correct values can be found in the left column. Abbreviation list: valid measures—N; count of patients exceeding cut-off point—n; standard deviation—SD; body mass index—BMI; transient ischemic attack—TIA; chronic obstructive pulmonary disease—COPD; low- vs. medium-risk groups—^a^; low- vs. high-risk groups—^b^; and medium- vs. high-risk groups—^c^.

**Table 3 microorganisms-12-01238-t003:** Applied treatment and necessary procedures in the study and control groups after C_2_HEST risk stratification.

Variables, Units	All Pts		Low Risk [0–1]		Medium [2–3]		High Risk [≥4]		*p*-Value		Post Hoc Analysis for Significant *p*	
	No COPD	COPD n = 75	No COPD	COPD n = 6	No COPD	COPD n = 25	No COPD	COPD n = 44	No COPD	COPD	No COPD	COPD
**Treatment and procedures administered**												
**The highest level of respiratory assistance provided throughout the hospital stay**												
No oxygen N, n (%)	1014 (48.1)	19 (25.3)	739 (52.3)	3 (50.0)	196 (42.0)	6 (24.0)	79 (34.3)	10 (22.7)	<0.0001	0.32	<0.0001 ^a^ 0.0008 ^b^ 0.53 ^c^	
Low flow oxygen support N, n (%)	724 (34.3)	39 (52.0)	448 (31.8)	3 (50.0)	178 (38.2)	10 (40.0)	98 (42.6)	26 (59.1)				
High-flow nasal cannula, Non-invasive ventilation N, n (%)	161 (7.6)	13 (15.0)	82 (5.8)	0 (0.0)	47 (11.8)	5 (20.0)	32 (13.9)	7 (15.9)				
Invasive ventilation N, n (%)	207 (9.8)	5 (6.7)	141 (10.0)	0 (0.0)	45 (9.7)	4 (16.0)	21 (9.1)	1 (2.3)				
Oxygenation parameters from the period of qualification for advanced respiratory support: PaO2, mmHg Mean ± SD (min.–max.)	68.07 ± 25.32 (29–168) N = 150	104 ± 22.6 (74–125) n = 23	66.1 ± 25.1 (34–168)	N/A	69.7 ± 24.9 (29–130)	105.3 ± 827.4 (74–125) n = 11	76.5 ± 25.5 (38–137)	100	0.26	N/A		
Therapy with catecholamines, N, n (%)	207 (9.8)	11 (14.7)	131 (9.3)	0 (0.0)	41 (8.8)	4 (16.0)	35 (15.2)	7 (15.9)	0.014	0.78	1 ^a^ 0.024 ^b^ 0.045 ^c^	
Coronary revascularization or/and an indication for coronary revascularization, N, n (%)	22 (1.0)	4 (5.3)	8 (0.6)	0 (0.0)	8 (1.7)	3 (12.0)	6 (2.6)	2 (4.5)	0.006	0.27	0.048 ^a^ 0.0499 ^b^ 1 ^c^	
Hemodialysis, N, n (%)	70 (3.3)	2 (2.7)	47 (3.3)	0 (0.0)	12 (2.6)	1 (4.0)	11 (4.8)	1 (2.3)	0.31	0.99		
Systemic corticosteroids N, n (%)	1047 (49.6)	49 (65.3)	704 (49.9)	4 (66.7)	229 (49.0)	17 (68.0)	114 (49.6)	28 (63.6)	0.95	0.93		
Plasma of the recovered, N, n (%)	231 (11.0)	8 (10.7)	166 (11.8)	1 (16.7)	37 (7.9)	4 (16.0)	28 (12.2)	3 (6.8)	0.058	0.33		
Remdesivir, N, n (%)	327 (15.5)	16 (21.3)	235 (16.6)	1 (16.7)	65 (13.9)	7 (28.0)	27 (11.7)	8 (18.2)	0.09	0.68		
Antibiotics, N, n (%)	1183 (56.1)	58 (77.3)	743 (52.6)	4 (66.7)	284 (60.8)	19 (76.0)	156 (67.8)	35 (79.5)	<0.0001	0.69	0.007 ^a^ <0.0001 ^b^ 0.26 ^c^	

Presentation of variables: continuous mean ± standard deviation, results distribution (min.–max.), and the number of present values; the values are presented as numbers with a percentage. Information regarding the numbers showing the correct values can be found in the left column. List of used abbreviations: valid measurements—N; the number of patients with a parameter above the cut-off point—n; standard deviation—SD; body mass index—BMI; transient ischemic attack—TIA; chronic obstructive pulmonary disease—COPD; non-applicable—N/A; low- vs. medium-risk stratum—^a^; low- vs. high-risk stratum—^b^; and medium- vs. high-risk stratum—^c^.

**Table 4 microorganisms-12-01238-t004:** Total, in-hospital, and after-discharge (until 20 September 2021) all-cause mortality in the study and control groups according to C_2_HEST risk strata.

Variables, Units	All Pts n = 75		Low Risk [0–1] n = 6		Medium [2–3] n = 25		High Risk [≥4] n = 44		*p*-Value		Post Hoc Analysis for Significant *p*	
	No COPD	COPD n = 75	No COPD	COPD n = 6	No COPD	COPD n = 25	No COPD	COPD n = 44	No COPD	COPD	No COPD	COPD
**All-cause mortality rate**												
In-hospital mortality, n (%) n = 75	308 (14.6)	18 (24.0)	118 (8.4)	1 (16.7)	104 (22.3)	6 (24.0)	86 (37.4)	11 (25.0)	<0.0001	1	<0.0001 ^a,b,c^	
3-month mortality, n (%)	507 (24.0)	39 (54.2)	198 (14.0)	3 (60.0)	187 (40.0)	11 (44.0)	122 (53.0)	25 (59.5)	<0.0001	0.4578	<0.0001 ^a,b^ 0.012 ^c^	
6-month mortality, n (%)	536 (25.4)	42 (73.7)	211 (14.9)	3 (50.0)	196 (42.0)	12 (48.0)	129 (56.1)	27 (61.4)	<0.0001	0.3928	<0.0001 ^a,b^ 0.17 ^c^	
Mortality until 20.09.2021	556 (26.4)	42 (56.0)	216 (15.3)	3 (50.0)	204 (43.7)	12 (48.0)	136 (59.1)	27 (61.4)	<0.0001	0.5886	<0.0001 ^a,b^ 0.0005 ^c^	
**Hospitalization**												
Duration of hospitalization, days n = 75	12.4 ± 14.2 1–131	14.3 ± 13.6 1–72	11.6 ± 14.0 1–131	10.8 ± 19.5 1–50	13.0 ± 13.5 1–124	15.8 ± 15.2 1–72	16.5 ± 16.5 1–121	14.0 ± 11.9 1–46	<0.0001	0.8002	0.109 ^a^ <0.0001 ^b^ 0.017 ^c^	
End of hospitalization, n (%) Death	308 (14.6)	18 (24)	118 (8.4)	1 (16.7)	104 (22.3)	6 (24.0)	86 (37.4)	11 (25.0)	<0.0001	0.9476	<0.0001 ^a^ <0.0001 ^b^ 0.0013 ^c^	
Discharge to home—full recovery	1289 (61.1)	27 (36.0)	991 (70.2)	2 (33.3)	212 (45.4)	8 (32.0)	86 (37.4)	17 (38.6)				
Transfer to another hospital—worsening)	267 (12.7)	13 (17.3)	137 (9.7)	2 (33.3)	93 (19.9)	4 (16.0)	37 (16.1)	7 (15.9)				
Transfer to another hospital—in recovery	245 (11.6)	17 (22.7)	166 (11.8)	1 (16.7)	58 (12.4)	7 (28.0)	21 (9.1)	9 (20.5)				

Presentation of variables: continuous mean ± standard deviation, results distribution (min.–max.), and the number of present values; the values are presented as numbers with a percentage. Information regarding the numbers showing the correct values can be found in the left column. Abbreviation list: valid measures—N; patients exceeding cut-off point—n; standard deviation—SD; body mass index—BMI; transient ischemic attack—TIA; chronic obstructive pulmonary disease—COPD; low- vs. medium-risk groups—^a^; low- vs. high-risk groups—^b^; and medium- vs. high-risk groups—^c^.

**Table 5 microorganisms-12-01238-t005:** The hazard ratios for overall mortality across C_2_HEST risk stratification in both COPD and non-COPD groups.

		COPD			Non-COPD	
Total Deaths	HR	95% CI	*p*-Value	HR	95% CI	*p*-Value
Overall	1.04	0.88–1.23	0.64	1.42	1.37–1.48	<0.0001
**Risk strata**						
Medium- vs. low-risk	0.84	024–2.99	0.79	3.44	2.84–4.16	<0.0001
High- vs. low-risk	1.15	0.35–3.80	0.81	5.11	4.12–6.34	<0.0001

**Table 6 microorganisms-12-01238-t006:** The hazard ratios for all-cause mortality during hospitalization across C_2_HEST risk stratification in both COPD and non-COPD groups (N/A—not applicable).

		COPD			Non-COPD	
In-Hospital Deaths	HR	95% CI	*p*-Value	HR	95% CI	*p*-Value
Overall	1.12	0.85–1.47	0.43	1.28	1.21–1.36	<0.0001
**Risk strata**						
Medium- vs. low-risk	0.96	0.11–8.28	0.97	2.34	N/A	N/A
High- vs. low-risk	1.02	0.13–8.14	0.98	3.05	N/A	N/A

**Table 7 microorganisms-12-01238-t007:** Clinical non-fatal events and hospitalization outcomes in the C_2_HEST risk strata in the COPD and non-COPD cohorts.

Variables, Units	All Pts n = 75		Low Risk [0–1] n = 6		Medium [2–3] n = 25		High Risk [≥4] n = 44		*p*-Value		Post Hoc Analysis for Significant *p*	
Selected Comorbidities	No COPD	COPD n = 75	No COPD	COPD n = 6	No COPD	COPD n = 25	No COPD	COPD n = 44	No COPD	COPD	No COPD	COPD
Aborted cardiac arrest, n (%) n = 75	23 (1.1)	1 (1.3)	15 (1.1)	0 (0.0)	3 (0.6)	0 (0.0)	5 (2.2)	1 (2.3)	0.19	1		
Shock, n (%) n = 75	181 (8.6)	7 (9.3)	108 (7.6)	1 (16.7)	42 (9.0)	4 (16.0)	31 (13.5)	2 (4.5)	0.013	0.207961	1 ^a^ 0.015 ^b^ 0.28 ^c^	
Hypovolemic shock, n (%) n = 75	33 (1.6)	2 (2.7)	22 (1.6)	0 (0.0)	5 (1.1)	2 (8.0)	6 (2.6)	0	0.29	0.262703		
Cardiogenic shock, n (%) n = 75	27 (1.3)	5 (6.7)	6 (0.4)	1 (16.7)	9 (1.9)	2 (8.0)	12 (5.2)	2 (4.5)	<0.0001	0.331994	0.012 ^a^ <0.0001 ^b^ 0.09 ^c^	
Septic shock, n (%) n = 75	137 (6.5)	4 (5.3)	89 (6.3)	0	28 (6.0)	2 (8.0)	20 (8.7)	2 (4.5)	0.348	0.727591		
Venous thromboembolic disease, n (%) n = 75	67 (3.1)	2 (2.7)	47 (3.3)	0 (0.0)	13 (2.8)	0 (0.0)	7 (3.0)	2 (4.5)	0.83	0.603604		
Pulmonary embolism, n (%) n = 75	47 (2.2)	1 (1.3)	39 (2.7)	0 (0.0)	11 (2.3)	0 (0.0)	6 (2.6)	1 (2.3)	0.98			
Deep vein thrombosis, n (%) n = 75	20 (0.9)	1 (1.3)	15 (1.1)	0 (0.0)	4 (0.9)	0 (0.0)	1 (0.4)	1 (2.3)				
Myocardial infarction, n (%) n = 75	26 (1.2)	0 (0.0)	8 (0.6)	0 (0.0)	10 (2.1)	0 (0.0)	8 (3.5)	0 (0.0)	0.0001	<0.0001	0.015 ^a^ 0.0018 ^b^ 0.95 ^c^	<0.0001 ^a,b^ 0.0665 ^c^
Myocardial injury, n (%) 3x n = 53	276 (24.6)	22 (41.5) n = 53	112 (16.5)	1 (33.3) n = 3	91 (31.6)	7 (41.2) n = 17	73 (46.2)	14 (42.4) n = 33	<0.0001	0.999999	<0.0001 ^a^ <0.0001 ^b^ 0.009 ^c^	
Myocardial injury, n (%) 5x n = 53	207 (18.5)	18 (34.0) n = 53	89 (13.2)	1 (33.3) n = 3	66 (22.9)	7 (41.2) n = 89	52 (32.9)	10 (30.3) n = 87	<0.0001	0.786862	0.0007 ^a^ <0.0001 ^b^ 0.09 ^c^	
Acute heart failure, n (%) n = 75	72 (3.4)	4 (5.3)	8 (0.6)	0 (0.0)	22 (4.7)	0 (0.0)	42 (18.3)	4 (9.1)	<0.0001	0.37735	<0.0001 ^a,b,c^	
Stroke/TIA, n (%) n = 75	43 (2.0)	1 (1.3)	18 (1.3)	0 (0.0)	19 (4.1)	0 (0.0)	6 (2.6)	1 (2.3)	0.0012	1	0.002 ^a^ 0.4 ^b^ 1 ^c^	
New cognitive signs and symptoms, n (%) n = 75	117 (5.5)	4 (5.3)	37 (2.6)	1 (16.7)	51 (10.9)	0 (0.0)	29 (12.6)	3 (6.8)	<0.0001	0.182401	<0.0001 ^a,b^ 1	
Pneumonia, n (%) n = 75	1009 (47.8)	52 (69.3)	602 (42.6)	4 (66.7)	265 (56.7)	14 (56.0)	142 (61.7)	34 (77.3)	<0.0001	0.1817010	<0.0001 ^a,b^ 0.72 ^c^	
Complete respiratory failure, n (%) n = 20	134 (6.3)	12 (60.0) n = 20	56 (4.0)	1 (100.0) n = 1	42 (9.0)	4 (66.7) n = 6	36 (15.7)	7 (53.8) n = 13	0.049	0.99999	1 ^a^ 0.068 ^b^ 0.33 ^c^	
SIRS, n (%) n = 75	210 (10.3)	10 (13.3)	140 (10.4)	2 (33.3)	40 (8.6)	2 (8.0)	30 (13.1)	6 (13.6)	0.18	0.254624		
Sepsis, n (%) n = 24	21 (2.4)	2 (8.3) n = 24	9 (1.6)		7 (1.5)	0 n = 6	5 (2.1)	2 (11.1) n = 18	0.037	0.254624	0.2119 ^a^ 0.16 ^b^ 1 ^c^	
Acute kidney injury, n (%) n = 75	223 (10.5)	14 (18.7)	111 (7.9)	0 (0.0)	62 (13.3)	5 (20.0)	50 (21.7)	9 (20.5)	<0.0001	0.730629	0.002 ^a^ <0.0001 ^b^ 0.018 ^c^	
Acute liver dysfunction, n (%) n = 69	65 (3.4)	1 (1.4) n = 69	30 (2.4)	0 (0.0) n = 6	22 (5.0)	0 (0.0) n = 25	13 (6.0)	1 (2.63) n = 38	0.0027	0.999999	0.03 ^a^ 0.02 ^b^ 1 ^c^	
Multiple organ dysfunction syndrome, n (%) n = 75	35 (1.7)	2 (2.7)	20 (1.4)	1 (16.7)	7 (1.5)	1 (4.0)	8 (3.5)	0 (0.0)	0.09	0.059459		
Lactic acidosis (on admission) n = 17	20 (8.7)	2 (11.8) n = 17	9 (8.7)	0 (0.0) n = 1	5 (6.8)	0 (0.0) n = 5	6 (12.0)	2 (18.2) n = 11	0.59	1		
Hyperlactatemia (on admission) n = 17	158 (69.3)	9 (52.9) n = 17	77 (74.0)	1 (100.0) n = 1	49 (66.2)	3 (60.0) n = 5	32 (64.0)	5 (45.5) n = 11	0.35	0.999999		
Bleedings, n (%) n = 75	110 (5.2)	4 (5.3)	64 (4.5)	0 (0.0)	24 (5.1)	1 (4.0)	22 (9.6)	3 (6.8)	0.006	0.999999	1 ^a^ 0.008 ^b^ 0.12 ^c^	
Intracranial bleeding, n (%) n = 75	21 (1.0)	0 (0.0)	12 (0.8)	0 (0.0)	8 (1.7)	0 (0.0)	1 (0.4)	0 (0.0)	0.205	<0.0001		<0.0001 ^a,b^ 0.0665 ^c^
Respiratory tract bleeding, n (%) n = 75	33 (1.6)	1 (1.3)	23 (1.6)	0 (0.0)	4 (0.9)	0 (0.0)	6 (2.6)	1 (2.3)	0.2	1		
Gastrointestinal tract bleeding, n (%) n = 75	39 (1.8)	2 (2.7)	20 (1.4)	0 (0.0)	9 (1.9)	0 (0.0)	10 (4.3)	2 (4.5)	0.029	0.603604	1 ^a^ 0.02 ^b^ 0.4 ^c^	
Urinary tract bleeding, n (%) n = 75	17 (0.8)	1 (1.3)	9 (0.6)	0 (0.0)	3 (0.6)	1 (4.0)	5 (2.2)	0 (0.0)	0.08	0.413333		

Presentation of variables: continuous mean ± standard deviation, results distribution (min.–max.), and the number of present values; the values are presented as numbers with a percentage. Information regarding the numbers showing the correct values can be found in the left column. Abbreviation list: valid measurements—N; number of patients exceeding the cut-off point—n; standard deviation—SD; body mass index—BMI; transient ischemic attack—TIA; chronic obstructive pulmonary disease—COPD; low- vs. medium-risk category—^a^; low- vs. high-risk category—^b^; and medium- vs. high-risk category—^c^.

## Data Availability

The datasets used and/or analyzed during the present study are available in this paper and in the Appendix A.

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
