# Peer review of "The Usefulness of the C2HEST Score in Predicting the Clinical Outcomes of COVID-19 in COPD and Non-COPD Cohorts"

_microorganisms, 2024, doi:10.3390/microorganisms12061238_

Round 1

Reviewer 1 Report

Comments and Suggestions for Authors

In the current study, the authors have checked the suitability of the C2HEST score to predict the clinical outcome in patients with COPD patients infected with COVID-19 where they demonstrated that the predictive ability of that scale is much lower in COPD cases. While the current study covers an important point of research, it needs many improvements, especially in language and data presentation. I have the following comments to the authors:

1. Why did the authors write their abstract in two paragraphs? 

2. In line 38, replace (the study) with (the current study) or (our study).

3. Add a reference to the second sentence of the first paragraph of the introduction. 

4. There are several typos such as (sparticipating) in line 114. Kindly revise the whole manuscript for these typos.

5. Regarding tables 2, 3, 4, and 7, the last column is not clear.

6. Make a separate file for supplementary data. Do not add supplementary tables in the manuscript. 

7. You put tables 1:7 under the supplementary material? Could you explain?

8. Is it C2HEST or C2HEST? Use one format throughout the whole manuscript.

Comments on the Quality of English Language

Moderate editing of the English language required

Author Response

Dear Reviewer,

We would like to express our sincere gratitude for your thorough review and valuable feedback on our manuscript titled “The usefulness of the C2HEST Score in Predicting the Clinical Outcomes of COVID-19 in COPD and Non-COPD Cohorts” (Manuscript ID: microorganisms-3036494). We have carefully considered your comments and suggestions and have made the necessary revisions to improve the quality and clarity of our work. Below, we provide a detailed response to each of your comments.

1) We decided that it was worth highlighting further information contained in the abstract by placing it in a separate paragraph. In our view, it was appropriate to divide the abstract into information regarding the main concept of the study and then describe the details concerning the characteristics of the studied population. Nevertheless, these are not issues we will insist on. Therefore, following the suggestion, we have combined the information into one paragraph.

2) We have made the relevant change as suggested.

3) The second sentence in the first paragraph of the Introduction, namely: “According to the data from Johns Hopkins University of Medicine on Oct 3rd, 2023, there have been more than 676 million infections since the beginning of the pandemic, of which almost 6.9 million have ended in death,” is marked with the appropriate reference. Please confirm if this is the sentence in question.

4) The paper has been checked, and typos have been corrected.

5) The tables mentioned by you have been formatted to fit entirely on the page.

6) As suggested, we will prepare the Supplementary Tables in a separate file.

7) We mistakenly assumed that all tables should belong to the Supplementary Material. Tables 1-7 have been excluded from this list.

8) We have standardized to the "C2HEST" version.

Once again, we would like to thank you for your valuable comments and suggestions. We believe that the revisions we have made have significantly improved the quality of our manuscript. We hope that our responses and the revised manuscript meet your expectations.

Sincerely,

Olgierd Dróżdż

Reviewer 2 Report

Comments and Suggestions for Authors

The authors present an analysis of the usefulness of the C2HEST Score in Predicting the Clinical Outcomes of COVID-19 in COPD and Non-COPD Cohorts. The paper analyses 2184 COVID-19 patients hospitalized at the University Hospital in Wroclaw between February 2020 and June 2021, split between 75 patients with COPD and 2109 patients without COPD.

This paper - as currently written - does not do a good job of highlighting what is new, and what has been previously published, as this is a reanalysis of a pre-existing dataset. In fact, the authors have previously published an analysis of this identical cohort in the following manuscript.

Usefulness of C2HEST Score in Predicting Clinical Outcomes of COVID-19 in Heart Failure and Non-Heart-Failure Cohorts: PMID: 35743564

The re-used text is very substantial (with light rewriting to reduce the iThenticate score), and in my view, the analysis is just a repeat of the workflow but applied to a COPD split instead of a heart failure split.

It is furthermore inadequate in my view that the previous analysis is only cited twice (as far as I can tell), in lines 83 and 427. This does not acknowledge the previous work sufficiently.

This does not mean that the COPD manuscript cannot be considered for publication, or that it does not have merit. But certainly, the manuscript would have to be very substantially rewritten, as follows:

1.       The Abstract, Introduction and Discussion should make it clear that this is a reanalysis of a dataset that the authors have already published.

2.       Large sections can be made more concise, for example under Materials and Methods, the authors can simply state "Study design and participants were as published in Rola et al (2022); the work presented here is a reanalysis of the data analysed there".

3.       In addition, the authors in my view should update the analysis as a good deal has changed since the cohort analysed was recruited. For example, new variants have emerged, with substantially different symptoms and presentation as compared with wildtype COVID-19 and its impact on a naive population. See for example PMID: 36005585, PMID: 36005585, and PMID: 36005585, which will help to provide context that the symptoms and biological impact of COVID-19 have changed as time has moved on.

Should the manuscript be substantially condensed so that the novel information only is emphasised, and additional context added so that the reader will be aware that findings about a C2HEST Score from 2020-2021 may not translate into C2HEST Scores in later waves, including present clinical practice, I do not see why it would not be suitable for publication. But the authors in my view have presented this article as a substantial new piece of work, when the workflow is identical, the population identical, and the majority of the conclusions about the non-COPD cohort are identical to the previous conclusions about the non-heart failure cohort.

I also note that the authors have presented several other analyses of the identical material, for example “Usefulness of the C(2)HEST Score in Predicting the Clinical Outcomes of COVID-19 in Diabetic and Non-Diabetic Cohorts.” (PMID: 35160324) and “The Usefulness of the C(2)HEST Risk Score in Predicting Clinical Outcomes among Hospitalized Subjects with COVID-19 and Coronary Artery Disease.” (PMID: 36016394). Again, to be clear, there is nothing wrong with reanalysis, such as re-examining a dataset with a different split of the data. I am sure that what the authors have done is inadvertent, but it is not good scientific behaviour to present a reanalysis of a previously published dataset without acknowledging the previous dataset and making clear through thorough referencing what is novel about the latest work, and what is not novel. 

Comments on the Quality of English Language

No comments

Author Response

Dear Reviewer,

We would like to express our sincere gratitude for your thorough review and valuable feedback on our manuscript titled “The usefulness of the C2HEST Score in Predicting the Clinical Outcomes of COVID-19 in COPD and Non-COPD Cohorts” (Manuscript ID: microorganisms-3036494). We have carefully considered your comments and suggestions and have made the necessary revisions to improve the quality and clarity of our work. Below, we provide a detailed response to each of your comments.

1) It is true that the group of 2184 individuals was already used in the study you mentioned. However, only the general study group is the same; the division into subpopulations of patients with COPD and those without COPD is unique to this study. Nevertheless, in accordance with your suggestion, we have added appropriate sections to the Abstract, Introduction, and Discussion so that future readers will have a clear understanding of the situation.

2) As noted above, the study design and the general group of 2184 patients are the same as in the work by Rola et al. (2022), which we have emphasized by adding appropriate sections. The division into COPD and non-COPD groups, which is unique to this study, was made based on GOLD criteria, distinguishing this study from the similar one concerning heart failure. Therefore, the sentence you suggested has been added to the manuscript. However, we believe that further descriptions regarding the collection of nasopharyngeal samples and the classification of COPD patients according to GOLD criteria are essential for future readers to fully understand the course and assumptions of our study.

3) We agree with your opinion on the need to update the data – indeed, since 2021, much has changed regarding virus mutations and the gradual immunization of the population through vaccinations. This should be the subject of future studies, which will undoubtedly provide interesting results. However, our study is retrospective and evaluates a population from a specific period in 2020-2021, a crucial time considering these were the early months of the pandemic, filled with uncertainties for global medicine. Therefore, to maintain the retrospective nature of the study, we do not include information updating the data for later years.

Once again, we would like to thank you for your valuable comments and suggestions. We believe that the revisions we have made have significantly improved the quality of our manuscript. We hope that our responses and the revised manuscript meet your expectations.

Sincerely,

Olgierd Dróżdż

Reviewer 3 Report

Comments and Suggestions for Authors

Dear authors,

I have now completed the review of the manuscript titled "The usefulness of the C2HEST Score in Predicting the Clinical Outcomes of COVID-19 in COPD and Non-COPD Cohorts."

In the present study, authors provide an initial evaluation of the C2HEST score in COPD patients with COVID-19. The contrast with the stronger predictive ability in the non-COPD cohort is an interesting finding worth additional exploration.

The manuscript is interesting and, in general, fairly well-written.

I have some suggestions to further improve the quality of the manuscript.

I would like to suggest that the authors address these limitations in the article, either by discussing them in the limitations section or, where feasible, by making the appropriate revisions:

1. The introduction need to be expanded. Current description does not represent multi faceted COVID-19, especially recent result like Long-term autoimmune inflammatory rheumatic outcomes of COVID-19, and COVID-19 susceptibility and clinical outcomes in inflammatory bowel disease. Moreover, clinicians may think Prevalence and Mortality Risk of Neurological Disorders during the COVID-19 Pandemic is also important.

2. Discussion on vaccination is lacking. In recent Comprehensive Systematic Reviews, Immunogenicity of COVID-19 Vaccines in Patients with Diverse Health Conditions are dealt. Vaccine can react differently on  diverse patient, and this should be discussed.

3. More detail needed on COPD diagnosis and severity. The methods state COPD was diagnosed according to GOLD criteria, but more specifics on the diagnostic criteria used and the severity distribution of COPD in the cohort would be informative for interpreting the results. COPD severity could impact COVID-19 outcomes.

4. Predictive ability of C2HEST appears poor in COPD. While the authors aimed to evaluate the utility of this predictive score in COPD patients, the results suggest very limited predictive value in this population, with no significant association with mortality outcomes. More research is likely needed to identify predictive tools better suited for COPD patients with COVID-19.

Thank you for your valuable contributions to our field of research. I look forward to receiving the revised manuscript.

Author Response

Dear Reviewer,

We would like to express our sincere gratitude for your thorough review and valuable feedback on our manuscript titled “The usefulness of the C2HEST Score in Predicting the Clinical Outcomes of COVID-19 in COPD and Non-COPD Cohorts” (Manuscript ID: microorganisms-3036494). We have carefully considered your comments and suggestions and have made the necessary revisions to improve the quality and clarity of our work. Below, we provide a detailed response to each of your comments.

1) The data presented in our study pertain to the years 2020-2021, a period during which the long-term consequences of SARS-CoV-2 infection were not yet known. It is true that we now observe post-COVID complications, likely of an autoimmune nature, which require further investigation. Nevertheless, following your suggestion, we decided to add an appropriate section on long COVID to illustrate the diverse problems that COVID-19 continues to pose.

2) The years 2020-2021 mark the early period of the pandemic when vaccines were initially unavailable, and later there were issues with their accessibility. Therefore, our database does not contain information about vaccinations. However, we agree that vaccinating patients could have impacted our study results. We have added a relevant note on this topic in the Limitations section.

3) The diagnosis of COPD was based on GOLD criteria, according to which patients were classified into either the COPD or non-COPD group based on patient history and medical records. No further subdivisions were made. Subsequent studies did not focus on distinguishing the severity groups of COPD but rather on scoring these patients using the C2HEST scale.

4) Our study indeed aimed to evaluate the utility of the C2HEST score in predicting various outcomes in COPD patients with COVID-19. While it is true that the predictive ability of the C2HEST score for mortality specifically in COPD patients was found to be lower, our findings indicate that the score is still valuable in this context. Our results demonstrated that the C2HEST score is effective in predicting the risk of death and various complications, such as the need for intensive care and mechanical ventilation, in the broader population of SARS-CoV-2 infected patients. Within the COPD subgroup, although the overall predictive power for mortality was limited, the C2HEST score showed significant utility in identifying patients at particularly high risk of cardiac complications during COVID-19. This suggests that while the score may not be as strong a predictor of mortality in COPD patients, it remains a useful tool for stratifying risk and managing cardiac complications in this vulnerable population.

Once again, we would like to thank you for your valuable comments and suggestions. We believe that the revisions we have made have significantly improved the quality of our manuscript. We hope that our responses and the revised manuscript meet your expectations.

Sincerely,

Olgierd Dróżdż

Round 2

Reviewer 1 Report

Comments and Suggestions for Authors

The manuscript can be accepted in the current format

Comments on the Quality of English Language

 Minor editing of the English language required

Author Response

Dear Reviewer,

thank you for your valuable advice and positive feedback on our work.

Sincerely,

Olgierd Dróżdż

Reviewer 2 Report

Comments and Suggestions for Authors

Thank you to the authors for thoroughly updating the manuscript, especially in acknowledging that this is a reanalysis.

Line 492, probably "the first" would be better than "the initial"

Line 558, thank you for including a limitations paragraph, it would also be worth flagging that symptoms and metabolic alterations from COVID have changed with different variants. This does not invalidate the research (which would be relevant for new pandemics and new illnesses) but it is a limitation that the C2HEST scoring / conclusions here would possibly not be reproducible in a new cohort (references https://doi.org/10.3390/metabo12080713, https://doi.org/10.1101/2022.05.21.22275368)

The authors will appreciate that the learnings from this paper would be difficult to apply to the current endemic COVID-19 situation in the world where the virus circulates freely but major health programmes are no longer considered necessary or cost:effective, given the altered balance of risks to the population. Some minor rewording to stress that the learnings may still be applied to pandemic preparedness for the future might be helpful.

Other than this I have no further changes to suggest.

Author Response

Dear Reviewer,

thank you for your valuable advice and positive feedback on our work.

In the meantime, we had to revise the content of the article to pass the similarity report. As a result, the line numbers you refer to may differ from those in the current manuscript. Nevertheless, we have made an effort to locate the appropriate sections and implement the suggested corrections.

Sincerely,

Olgierd Dróżdż

Reviewer 3 Report

Comments and Suggestions for Authors

All comments have been thoroughly addressed. I extend my gratitude to both the authors and editors for taking my opinions into consideration during the review of this manuscript.

Author Response

(The authors gave the same response as above.)
